# Deep Learning and Registration-Based Mapping for Analyzing the Distribution of Nodal Metastases in Head and Neck Cancer Cohorts: Informing Optimal Radiotherapy Target Volume Design

**DOI:** 10.3390/cancers15184620

**Published:** 2023-09-18

**Authors:** Thomas Weissmann, Sina Mansoorian, Matthias Stefan May, Sebastian Lettmaier, Daniel Höfler, Lisa Deloch, Stefan Speer, Matthias Balk, Benjamin Frey, Udo S. Gaipl, Christoph Bert, Luitpold Valentin Distel, Franziska Walter, Claus Belka, Sabine Semrau, Heinrich Iro, Rainer Fietkau, Yixing Huang, Florian Putz

**Affiliations:** 1Department of Radiation Oncology, University Hospital Erlangen, Friedrich-Alexander-Universität Erlangen-Nürnberg, 91054 Erlangen, Germany; thomas.weissmann@uk-erlangen.de (T.W.); sebastian.lettmaier@uk-erlangen.de (S.L.); daniel.hoefler@uk-erlangen.de (D.H.); lisa.deloch@uk-erlangen.de (L.D.); stefan.speer@uk-erlangen.de (S.S.); benjamin.frey@uk-erlangen.de (B.F.); udo.gaipl@uk-erlangen.de (U.S.G.); christoph.bert@uk-erlangen.de (C.B.); luitpold.distel@uk-erlangen.de (L.V.D.); sabine.semrau@uk-erlangen.de (S.S.); rainer.fietkau@uk-erlangen.de (R.F.); 2Comprehensive Cancer Center Erlangen-EMN (CCC ER-EMN), 91054 Erlangen, Germany; matthias.may@uk-erlangen.de (M.S.M.); matthias.balk@uk-erlangen.de (M.B.); heinrich.iro@uk-erlangen.de (H.I.); 3Bavarian Cancer Research Center (BZKF), 81377 Munich, Germany; sina.mansoorian@med.uni-muenchen.de (S.M.); franziska.walter@med.uni-muenchen.de (F.W.); claus.belka@med.uni-muenchen.de (C.B.); 4Department of Radiation Oncology, University Hospital, Ludwig Maximilian University of Munich, 81377 Munich, Germany; 5Department of Radiology, University Hospital Erlangen, Friedrich-Alexander-Universität Erlangen-Nürnberg, 91054 Erlangen, Germany; 6Translational Radiobiology, Department of Radiation Oncology, University Hospital Erlangen, Friedrich-Alexander-Universität Erlangen-Nürnberg, 91054 Erlangen, Germany; 7Department of Otolaryngology, Head and Neck Surgery, Friedrich-Alexander-Universität Erlangen-Nürnberg, 91054 Erlangen, Germany

**Keywords:** deep learning, artificial intelligence, nodal metastasis, head and neck cancer, radiotherapy, target volume design, atlas, mapping, autosegmentation, lymph node level classification

## Abstract

**Simple Summary:**

This study presents two novel methods for automatically analyzing the distribution of nodal metastases in head and neck (H/N) cancer cohorts. The proposed deep learning method uses lymph node level autosegmentation to automatically assign lymph node metastases to 20 H/N nodal levels. The second, registration-based method maps lymph nodes into a calculated average-anatomy template CT, allowing for the analysis and visualization of the 3D probability distribution of metastases without predefined level boundaries. Both methods were evaluated on a cohort of 193 H/N cancer patients, with the deep learning method being able to correctly categorize all 449 lymph nodes to their respective levels as determined by a multireader expert review. Level 2 was by far the most frequently involved level (59% of lymph nodes). The mapping technique showed clustering in high-risk regions and proved to be consistent with the ground-truth distribution. Both methods could contribute to the refinement of H/N radiotherapy target volume design.

**Abstract:**

We introduce a deep-learning- and a registration-based method for automatically analyzing the spatial distribution of nodal metastases (LNs) in head and neck (H/N) cancer cohorts to inform radiotherapy (RT) target volume design. The two methods are evaluated in a cohort of 193 H/N patients/planning CTs with a total of 449 LNs. In the deep learning method, a previously developed nnU-Net 3D/2D ensemble model is used to autosegment 20 H/N levels, with each LN subsequently being algorithmically assigned to the closest-level autosegmentation. In the nonrigid-registration-based mapping method, LNs are mapped into a calculated template CT representing the cohort-average patient anatomy, and kernel density estimation is employed to estimate the underlying average 3D-LN probability distribution allowing for analysis and visualization without prespecified level definitions. Multireader assessment by three radio-oncologists with majority voting was used to evaluate the deep learning method and obtain the ground-truth distribution. For the mapping technique, the proportion of LNs predicted by the 3D probability distribution for each level was calculated and compared to the deep learning and ground-truth distributions. As determined by a multireader review with majority voting, the deep learning method correctly categorized all 449 LNs to their respective levels. Level 2 showed the highest LN involvement (59.0%). The level involvement predicted by the mapping technique was consistent with the ground-truth distribution (p for difference 0.915). Application of the proposed methods to multicenter cohorts with selected H/N tumor subtypes for informing optimal RT target volume design is promising.

## 1. Introduction

Radiotherapy serves as a crucial component in the treatment of patients with head and neck (H/N) squamous carcinoma, playing a pivotal role in both definitive and adjuvant contexts. Over the recent decades, technical advancements such as intensity-modulated radiation therapy (IMRT) and volumetric modulated arc therapy (VMAT) have enhanced tumor control and significantly reduced toxicity, ultimately leading to increased survival rates [1]. Nonetheless, the inherent short- and long-term side effects of head and neck radiotherapy underscore the importance of enhancing functional outcomes in H/N cancer treatments [2,3,4,5]. De-escalation strategies in H/N cancer present inherent risks due to the reduced salvage potential and adverse survival outcomes for recurrent tumors [6]. Various treatment-de-escalation approaches have been pursued over time, with unilateral radiation in the adjuvant setting becoming increasingly accepted and attempts at de-escalation in the definitive setting, including dose and volume reduction approaches, being proposed [7,8]. However, although dose and volume de-escalation is gaining traction for HPV-associated oropharyngeal cancer [9,10], similar de-escalation options in biologically less-favorable tumor types are not available.

Fundamentally, despite the precision enabled by advanced dose delivery and image guidance, a challenge that persists in head and neck radiotherapy is accurately targeting microscopic nodal involvement. The current elective neck treatment usually still involves the radiation of larger neck compartments with relatively homogeneous doses due to the inability to localize microscopic nodal disease through imaging. Advancing our comprehension of the spatial distribution of microscopic nodal involvement with novel image analysis methods is a promising approach to inform volume-reduction approaches and refine target volume design for elective neck radiotherapy.

Recent advances in image analysis including deep learning autosegmentation not only improve treatment planning efficiency [11,12] but also enable large-scale analyses of vast patient cohorts to inform optimal target volume design. The present work, therefore, introduces two methods for automatically analyzing the distribution of nodal metastases in large H/N cancer patient cohorts and makes the following key contributions:


We describe a deep learning method based on autosegmentation that automatically categorizes each nodal metastasis into a specific lymph node level based on its spatial proximity to autosegmented level boundaries. This method utilizes a previously described nnU-Net 3D/2D ensemble model to autosegment 20 head and neck levels [11] but extends it to this novel task using an algorithmic distance-based level assignment. We introduce a nonrigid-registration-based mapping method for H/N CT datasets that allows for the estimation, analysis, and visualization of the 3D probability distribution for the lymph node metastases of an entire patient cohort, independent of predefined level boundaries. Both methods are evaluated on a cohort of 193 head and neck cancer patient planning CTs including multireader expert assessment by three radiation oncologists demonstrating that the automated analysis of large head and neck patient cohorts for the purpose of improved nodal-level target volume design is feasible with high accuracy.


## 2. Materials and Methods

### 2.1. Patient Population and Dataset

Planning CT datasets from 193 patients undergoing definitive chemoradiation for H/N squamous carcinoma at the Department of Radiation Oncology of the University Hospital Erlangen from 01/2015 to 03/2021 were used for this analysis. Patients who had received systemic therapy for induction were excluded. Table 1 shows the characteristics of the cohort. All patients provided their written informed consent for their imaging data to be used for scientific investigations. Otherwise, no selection criteria were applied. IRB approval was not required for this study as per the institutional guidelines (institutional Ethics Committee of the University Hospital Erlangen) and the local legislation (Bavarian hospital law BayKrG Art. 27 (4)). Planning CTs were acquired using a thermoplastic mask (Softfix 5-point mask, Unger Medizintechnik GmbH, Mülheim-Kärlich, Germany) with an iodine contrast medium (Imeron 350, Bracco Imaging Deutschland Gmbh, Konstanz, Germany) being applied in the absence of contraindications. The CT scanners used for the planning CTs were Siemens SOMATOM go.Open Pro and Siemens Sensation Open. For all patients, the matrix size used was 512 × 512, and the slice thickness was 3 mm, which was the institutional standard for treatment planning in head and neck cancer. For each patient, the metastatic lymph nodes were identified and delineated for treatment planning by an experienced physician using contrast-enhanced CT imaging. In total, 449 lymph node metastases segmentations were contoured and exported from the treatment planning system.

### 2.2. Deep Learning Automated Analysis of the Distribution of Nodal Metastases 

To determine the nodal metastases distribution for the entire cohort in relation to the cervical lymph node levels, we exported the planning CT datasets and nodal GTVs from clinical routine in the DICOM-RT format from the treatment planning system. Initially, the datasets underwent preprocessing, and lymph node levels were autosegmented using an nnU-net 2D/3D ensemble model as previously reported [11]. The nnU-Net model differentiated 20 lymph node levels according to Grégoire et al., including level 1a, 6a, 6b, 7a and bilateral levels 1b, 2, 3, 4a, 4b, 5, 7b, and 8 [13].

The segmented datasets were then processed through an automated pipeline implemented in 3D Slicer (v. 4.11) [14]. The pipeline calculated the geometric center coordinate of each metastasis through an erosion operation to isolate individual nodes and a connected-component analysis. A lymph node was algorithmically assigned a specific level if its center coordinate was located within that level segmentation; otherwise, the nearest level segmentation was utilized by calculating the distance from the lymph node center to all level surfaces. Finally, the identified levels for all lymph node metastases across all datasets were combined to obtain the distribution for the entire cohort.

For all experiments, we used nnU-net version 1.6.6 [15], Python version 3.7.4, PyTorch version 1.9.0 [16] with CUDA version 11.1 [17], and CUDNN version 8.0.5. Model training, inference, and all computations were performed on a GPU workstation utilizing an Nvidia Quadro RTX 8000 with 48 GB of GPU memory (Santa Clara, CA, USA). We provide the pipeline program code and the model weights for download at the following location: https://github.com/putzfn/ (accessed on 18 September 2023).

### 2.3. Evaluation and Expert Review

For manual evaluation, three board-certified clinical experts (authors T.W., S.L. and F.P.) with 7, 19, and 11 years of experience in radiation oncology examined each lymph node metastasis using a dedicated workflow implemented in 3D Slicer. The workflow systematically displayed each nodal metastasis, planning CT dataset, and results from the deep learning pipeline to the expert. Figure 1 shows an example of how the predictions were presented to the experts for review during the assessment (font size increased for manuscript figure). The experts’ responsibility was to assess the correctness of the deep learning assignment and, in cases of disagreement, provide the correct level designation. Majority voting was used to combine the assessment of the three clinical reviewers for each individual lymph node and obtain the reference distribution for comparison.

### 2.4. Nonrigid-Registration-Based Mapping Analysis 

Complementary to the deep-learning-based method, we employed a nonrigid-registration-based mapping analysis to a common template. The mapping analysis allows for the estimation, analysis, and visualization of the 3D probability distribution for the lymph node metastases of the patient cohort, independent of predefined level boundaries. The mapping analysis is related to a method we previously described for analyzing the distribution of pelvic nodal metastases for prostate cancer CT datasets [18] but has been completely revised to allow for application in head and neck CT datasets including the construction of a cohort-average template dataset as a mapping target and the replacement of the registration framework.

To begin with, the average-anatomy template dataset was computed to establish an optimal mapping target for nonrigid-registration-based lymph node mapping, using the ANTs registration toolkit version 2.3.4 [19,20]. A subset of 32 randomly selected planning CT datasets from the cohort was employed for template calculation, as it is well established that using a representative subset in atlas generation maintains accuracy while being more computationally efficient [19]. The ANTs multivariate template construction pipeline, originally developed for brain MRI, was adapted for head and neck CT template construction. Briefly, CT datasets were initially rigidly co-registered, but linear registration was disabled during template construction; the voxel intensity summarization used the median instead of the mean; and the background voxel intensity was set to −1000. The template construction was carried out using BsplineSyn with the cross-correlation metric for 20 iterations, requiring 5 days on a dual CPU workstation (2x Intel Xeon Platinum 8260 with 96 GB of RAM, Santa Clara, CA, USA).

Following template construction, lymph node center coordinates from all 193 cases were mapped onto the common anatomy using nonrigid registration. Nonrigid-registration-based mapping was performed using ANTs registration with a sequence of rigid, affine, and BsplineSyn registration, with cost function masking applied to the primary tumor and nodal GTVs during registration.

Finally, kernel density estimation was employed to convert the mapped lymph node center locations into an estimate of the underlying average probability distribution for lymph node metastases. Three-dimensional kernel density estimation, based on the mapped lymph node center locations, was executed using the Python library KDEpy (Gaussian kernel, bandwidth 4, p-norm 2) [21]. A CT atlas and 3D renderings of the average distribution of head and neck nodal metastases for the whole cohort were generated with 3D Slicer 4.11 and Blender 3.5 [14,22]. To quantitatively validate the independent mapping technique, a head and neck radiotherapy expert with board certification manually delineated all lymph node levels for the average-anatomy template. Subsequently, the expected proportion of nodal metastases for every lymph node level was calculated from the estimated spatial probability distribution and compared to the ground-truth proportions with a chi-squared test (Figure 2). The program code for the mapping analysis and the calculated population-average template mapping target is also available for download at https://github.com/putzfn/ (accessed on 18 September 2023).

## 3. Results

### 3.1. Accuracy of the Deep Learning Method for the Nodal Metastases Distribution Analysis

A total of 193 patients with H/N squamous carcinoma, including 449 lymph node metastases, were part of the study. Table 2 and Figure 3 display the frequency of level involvement for the deep learning method in comparison with the expert correction and the independent mapping technique. Excluding export from the treatment planning system, computation for the whole 193-patient cohort by the deep learning method took 4 h and 40 min on a GPU workstation (average of 1 min 27 s per CT dataset). In total, 41 out of the 449 lymph node metastases (9.1%) were located outside the level autosegmentations and had been assigned based on their distance to the closest lymph node level. When combining the assessment by the three clinical reviewers via majority voting (each LN considered individually), the deep-learning-based method correctly categorized all 449 LNs (100%) to their respective levels. The assessment by the three clinical experts is shown in Table 2. For expert 1, 444 out of 449 lymph node metastases (98.9%) were determined to be correctly classified. Specifically, after expert review, two lymph node metastases initially assigned to level 2 were changed to level 3, one lymph node from level 8 to level 1b, one lymph node from level 8 to level 2, and one metastasis from right level 7b to level 7a. For expert 2, all lymph node metastases (100%) were assessed to be assigned correctly. Finally for expert 3, 448 out of 449 nodal metastases were correctly categorized (99.8%), and one lymph node metastases was changed from right level 2 to adjacent level 5.

### 3.2. Distribution of Nodal Metastases in Reference to Nodal Levels

The majority of metastases were located in level 2 (30.5% and 28.5% for right and left level 2, respectively), followed at a distance by lymph nodes in level 3 (11.4% and 11.6% for right and left level 3, respectively). Bilateral levels 1b and 5 also contained a significant portion of nodal metastases (3.8% for right and left level 1b, as well as 2.7% and 2.5% for right and left level 5). Lymph node metastases occurred infrequently in levels 4a, 6b, 7a, 7b, and 8. Notably, no lymph node metastases were observed in levels 1a, 4b, and 6a (Figure 3, Table 2).

### 3.3. Nonrigid-Registration-Based Mapping Analysis

The level involvement predicted by the spatial probability distribution of the mapping analysis was consistent with and not significantly different from the expert corrected and deep learning distribution (Figure 3, chi-squared p for difference 0.915). The independent mapping analysis showed a strong hotspot for nodal metastases in level 2 as well as a high probability for lymph node metastases in level 3. In the mapping analysis, nodal metastases were not evenly distributed throughout level volumes; instead, they predominantly clustered in specific hotspot regions along the carotid sheath (Figure 4).

## 4. Discussion

Analyzing large head and neck patient cohorts with novel image analysis methods is an important approach for improving radiotherapy target volume design by addressing microscopic nodal disease more precisely. In the present study, we introduce and evaluate a deep-learning- and a registration-based method for automatically analyzing the spatial distribution of nodal metastases in large head and neck cancer cohorts. As determined by multireader review with majority voting, the automated deep-learning-based method showed a very high accuracy of 100% and was able to identify the correct lymph node level for all 449 nodal metastases. Excluding dataset export from the treatment planning system, computation for the entire 193-patient cohort took just 4 h and 40 min for the deep-learning-based method. This corresponds to an average of 1 min and 27 s per dataset. Comparing the computation time of the automated method with the manual review time was outside the scope of this work. However, as the method is automated, no expert interaction time is required, and if expert validation of predicted classifications is desired, the calculations can be performed in advance. Moreover, the calculation time can be further accelerated by using newer GPU generations for deep learning autosegmentation. As the method scales linearly with the number of datasets, it is particularly suitable for large, multicenter datasets, where manual review alone would be either prohibitively costly or entirely unfeasible.

While deep learning autosegmentation is established for enhancing efficiency in radiotherapy treatment planning and autocontouring tasks in particular [11,23,24,25,26], the potential of automated methods for analyzing large patient cohorts and obtaining insights for target volume definition remains an emerging area of investigation. Thus far, artificial intelligence image analysis methods have predominantly been employed in image phenotyping; radiomic signature identification; and efforts to predict treatment responses, toxicities, and outcomes [27,28]. The task of deep learning lymph node level classification has been introduced more recently, and to the best of our knowledge, the present work is the first description of a deep-learning-based method for CT lymph node level classification in head and neck cancer. Iuga et al. recently described a deep learning method for classifying thoracic lymph node levels. Using a binary mask of semiautomatically delineated nodal metastases together with the thoracic CT volume as model input of a 3D foveal fully convolutional neural network (F-Net), the model was trained on voxel-wise classification of the nodes into 27 lymph node level classes. Interestingly, to determine the final class/level of each node, the authors selected the level with the maximum activation sum inside the lymph node mask. With this approach, a total thoracic lymph node level classification accuracy of 86.4% and a diagnostic N-staging accuracy of 96.14% were achieved [29]. Compared to the method by Iuga et al. of training a dedicated multiclass segmentation model for lymph node level classification, we make use of a preexisting model for the more frequent task of nodal-level segmentation in the present work. In this approach, the level classification for each lymph node is achieved by the predicted level voxel label at the lymph node center or by closest-level autosegmentation. This method may have the advantage that it does not require the training of an additional specialized segmentation model and, thus, may be easier to implement. Furthermore, since level autosegmentation is a well-researched task [11,30,31,32], advances in this field can, thereby, also be applied to lymph node level classification. However, to determine whether one method is superior in terms of classification accuracy, a direct comparison between these two deep learning strategies for lymph node level classification would be necessary.

The proposed method for the level classification of lymph node metastases uses a level autosegmentation model that was previously developed [11] and evaluated on an independent test set of 15 head and neck cases. Interestingly, this level autosegmentation model also showed high accuracy in the independent larger cohort of the present study for the related task of lymph node level classification, which could be seen as an indication for its generalizability. Nodal segmentations from clinical treatment planning were employed in the present study. However, it is worth noting that the presented method can also be easily extended by existing lymph node autosegmentation methods [33] if clinical segmentations are not available.

In addition to the deep-learning-based method, an independent mapping technique for head and neck cancer CT datasets was also introduced in this work. The nonrigid-registration-based mapping method allows for the estimation, analysis, and visualization of the 3D probability distribution for the lymph node metastases of an entire patient cohort, independent of prespecified level boundaries or autosegmentation methods. We previously described a simpler technique for pelvic prostate cancer datasets [18]. To allow for nonrigid-registration-based mapping in the anatomically variable head and neck region, among multiple optimizations, a cohort-average template dataset is constructed as optimal mapping target that represents the mean anatomy of the population and a high-performance BsplineSyn deformable registration was used (see methods section for full description). Notably, both the deep learning and nonrigid-registration-based techniques yielded consistent results, and the level involvement predicted by the mapping analysis was not significantly different from the expert corrected distribution. Although the nonrigid-registration-based mapping technique required a significantly longer computation time, it offers the advantage of analyzing lymph node distributions without relying on predefined level classifications. The advantage of not being constrained to prespecified level definitions could be particularly useful in identifying hypotheses for improved level boundary definitions that could later be backtested with the proposed deep-learning autosegmentation-based method. The mapping of nodal metastases and recurrences has a long-standing history in radiation oncology [34]. However, usually, mapping is performed manually by clinical experts with reference to identifiable anatomic structures [35,36,37,38,39,40], which is very time consuming and limits the feasibility of large-scale mapping studies. Therefore, the proposed automated, nonrigid-registration-based mapping method could help in accelerating research into optimal target volume design. As the population-average template dataset can be shared between multiple institutions without privacy constraints to allow for local mapping with a final pooling of the mapped lymph node coordinates, the method could be well suited for large multicenter collaborations. With this purpose in mind, we also share the population-average mapping target template calculated in the present study with the research community.

In the present cohort, head and neck lymph node metastases were identified via contrast-enhanced CT. Given that FDG PET has exhibited significantly higher specificity and numerically superior sensitivity compared to contrast-enhanced CT imaging in identifying metastatic lymph nodes in the head and neck [41], the application of the proposed image analysis techniques to extensive cohorts with FDG PET imaging holds particular promise. The planning CT datasets available for the present analysis had a slice thickness of 3 mm, which is in the range radiotherapy head and neck image analysis methods are currently developed and evaluated on [30,42]. A lower slice thickness will increase the resolution in the z-dimension but also the computational time for the mapping analysis.

To achieve a sufficiently large cohort size for evaluating the deep-learning- and registration-based mapping method, multiple head and neck tumor entities were included in the present study. Although the obtained distribution is representative for the entirety of head and neck squamous cancer cases treated with definitive chemoradiation at a tertiary care center, drawing conclusions for specific head and neck cancer subsets was outside the scope of the present work.

While the description and the evaluation of the two automated techniques for analyzing the distribution of nodal metastases were the primary objectives of this work, the obtained distribution of nodal metastases deserves discussion. In the present cohort, level 2 was most frequently affected, followed by level 3, which exhibited a notably lower frequency of involvement. Bilateral levels 1b and 5 also harbored a significant number of lymph nodes. Nodal metastases were infrequent in levels 4a, 6b, 7a, 7b, and 8, and no lymph node metastases were observed in levels 1a, 4b, and 6a. Studies examining the distribution of pathological lymph nodes primarily rely on older surgical series. Deo et al. reported a significant involvement of level 1 in oral cavity cancer, with lower levels being affected considerably less frequently [43]. Sanguineti et al. observed the highest pathological involvement of lymph nodes in levels 2, 3, and 4 for early-stage oropharyngeal carcinoma, while level 5 appeared to be affected significantly less often [44]. Candela et al. similarly found the highest involvement rates in levels 2, 3, and 4 for both oropharyngeal and hypopharyngeal primaries [45]. It is important to note that surgical distribution studies usually do not provide a more detailed subdivision within lymph node levels 1, 2, and 4, which could be relevant for volume reduction in patients undergoing radiotherapy.

Head and neck lymph node level definitions were originally developed for head and neck surgery as the anatomic definition of level compartments is needed to fulfill surgical requirements. Later, Grégoire et al. refined these surgical compartments for radiation oncology target volume design but kept the basic principles established in surgical practice [13]. Interestingly, the independent mapping technique in the present study revealed the nonuniform involvement of level volumes, with nodal metastases clustering in distinct hotspots, predominantly situated along the carotid sheath (Figure 4), which, to the best of our knowledge, has not been described previously. These findings could lead to the interesting hypothesis relevant for future studies that novel image analysis methods could help in further refining head and neck lymph node level definitions for radiotherapy treatment planning to allow for a more precise targeting of high-risk regions.

### Limitations

Limitations of the present study include the fact that lymph node metastases were identified via contrast-enhanced CT imaging. FDG PET-CT has higher specificity than contrast-enhanced CT imaging for detecting lymph node metastases [41] but was not available for the present analysis. Moreover, to allow for a sufficiently large cohort size, subgroup selection of biologically homogeneous head and neck tumor entities was not possible. Additionally, the primary tumor and pathologic lymph nodes had been manually delineated for treatment planning in the present analysis and autosegmentation of lymph node metastases was outside the scope of the present research. Applying the described methods to larger, multicenter cohorts with FDG PET-CT imaging and extending it to include lymph node autosegmentation will be the focus of our future research. Furthermore, we share all methods, program code, and the calculated population-average atlas of the present work with the same aim to the research community. Finally, the deep learning autosegmentation model employed in the present study was developed at the same institution on an independent dataset. Applying a segmentation model trained on a single institution dataset to a more heterogeneous multicenter study is expected to result in some performance decrease. In a multicenter context, it would, therefore, be optimal to train or finetune the autosegmentation model on a representative subset from all participating institutions.

## 5. Conclusions

In this study, we introduced and evaluated two novel methods for analyzing the distribution of lymph node metastases in a large cohort of head and neck cancer patients. The proposed deep learning method showed high accuracy and was able to assign all 449 lymph node metastases in the present cohort to their correct nodal levels, as determined by a multireader expert review with majority voting. The complementary nonrigid-registration-based mapping technique allows for the estimation of the 3D probability distribution for lymph node metastases of an entire patient cohort independent of predefined level boundaries. The mapping method revealed considerable inhomogeneity in the distribution of nodal metastases with clustering in particular high-risk regions and predicted level involvement consistent with the ground-truth distribution. Further application of the presented deep-learning- and mapping-based image analysis methods to evaluate the distribution of cervical lymph node metastases in large, multicenter cohorts could contribute to efforts in volume de-escalation and differentiated dose prescription in elective head and neck radiotherapy.

## Figures and Tables

**Figure 1 cancers-15-04620-f001:**
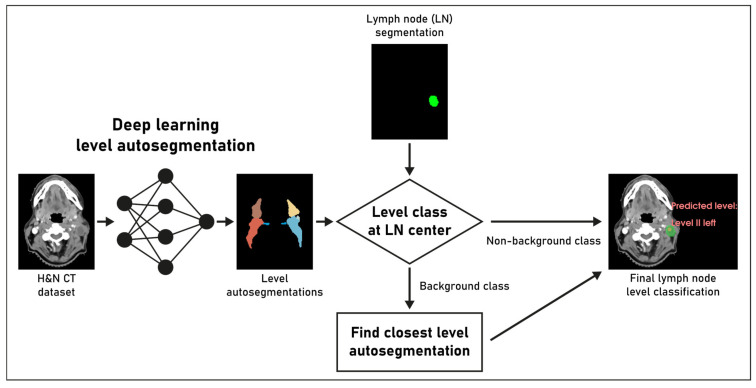
Overview of the proposed deep learning pipeline for lymph node (LN) level classification. See methods section. The final lymph node level classification on the right side of the image was presented to the clinical experts for review.

**Figure 2 cancers-15-04620-f002:**
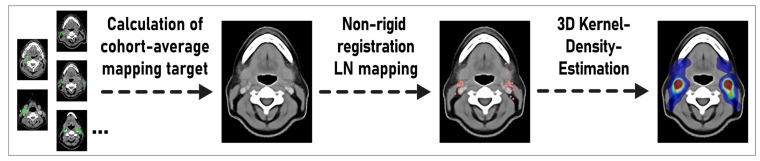
Overview of the proposed mapping method for estimating the probability density distribution of lymph node metastases in head and neck cancer cohorts independent from prespecified level boundary definitions. LN: lymph node.

**Figure 3 cancers-15-04620-f003:**
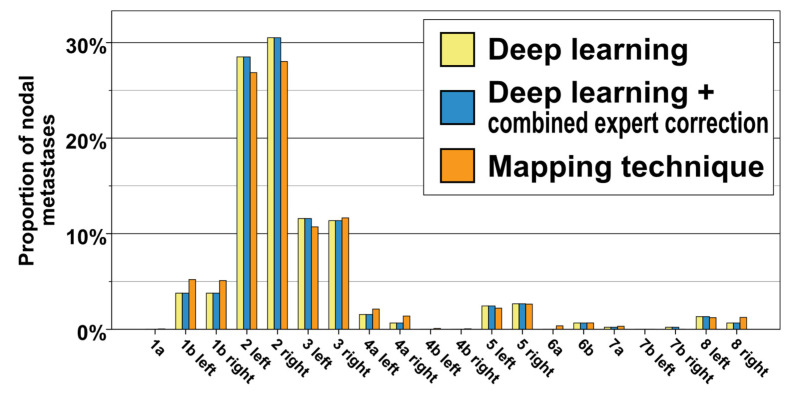
Distribution of nodal metastases for the whole cohort of *n* = 449 lymph node metastases and 193 patients. Yellow: deep learning method, blue: deep learning method + expert correction, orange: independent mapping technique.

**Figure 4 cancers-15-04620-f004:**
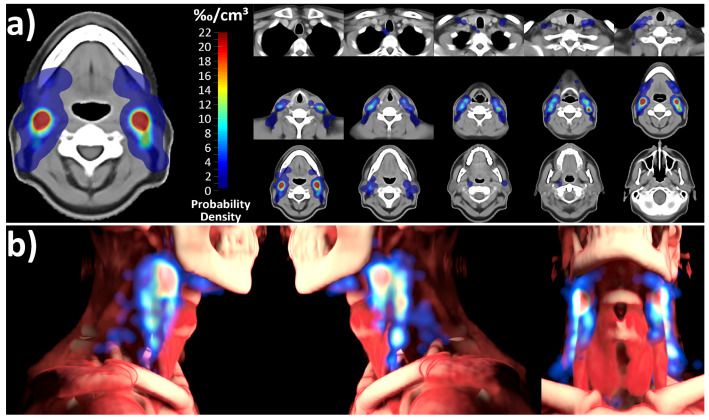
Probability density distribution of lymph node metastases for the entire cohort, determined using the nonrigid-registration-based mapping technique. A primary hotspot is evident in level 2, with additional significant hotspots in level 3. Notably, lymph node metastases are not uniformly distributed throughout level volumes but primarily cluster along the carotid sheath. (**a**) Atlas sections of the template anatomy overlaid with the calculated probability distribution. (**b**) 3D rendering of the probability distribution in relation to the average-anatomy template.

**Table 1 cancers-15-04620-t001:** Cohort characteristics.

Dataset Characteristic	Total Cohort (*N* = 193)
**Primary tumor location**	
Oropharynx	105 (54.4%)
Hypopharynx	42 (21.8%)
Larynx	19 (9.8%)
Nasopharynx	15 (7.8%)
Oral cavity	12 (6.2%)
**Primary tumor laterality**	
Left	88 (45.6%)
Right	91 (47.2%)
Bilateral	14 (7.3%)
**Primary tumor stage**	
T1	14 (7.3%)
T2	60 (31.1%)
T3	31 (16.1%)
T4	88 (45.6%)
**Nodal stage**	
N1	30 (15.5%)
N2	132 (68.4%)
N3	31 (16.1%)
**p16 status**	
Negative	149 (77.2%)
Positive	44 (22.8%)

**Table 2 cancers-15-04620-t002:** Frequency of level involvement for the whole cohort of *n* = 449 nodal metastases.

Lymph Node Level	Deep Learning	Expert Correction			
		Expert 1	Expert 2	Expert 3	Majority Voting
**Level 2 right**	**137 (30.5%)**	**136 (30.3%)**	137 (30.5%)	**136 (30.3%)**	137 (30.5%)
**Level 2 left**	128 (28.5%)	128 (28.5%)	128 (28.5%)	128 (28.5%)	128 (28.5%)
**Level 3 right**	**51 (11.4%)**	**53 (11.8%)**	51 (11.4%)	51 (11.4%)	51 (11.4%)
**Level 3 left**	52 (11.6%)	52 (11.6%)	52 (11.6%)	52 (11.6%)	52 (11.6%)
**Level 1b left**	**17 (3.8%)**	**18 (4.0%)**	17 (3.8%)	17 (3.8%)	17 (3.8%)
**Level 1b right**	17 (3.8%)	17 (3.8%)	17 (3.8%)	17 (3.8%)	17 (3.8%)
**Level 5 right**	12 (2.7%)	12 (2.7%)	12 (2.7%)	**13 (2.9%)**	12 (2.7%)
**Level 5 left**	11 (2.5%)	11 (2.5%)	11 (2.5%)	11 (2.5%)	11 (2.5%)
**Level 4a left**	7 (1.6%)	7 (1.6%)	7 (1.6%)	7 (1.6%)	7 (1.6%)
**Level 8 left**	**6 (1.3%)**	**5 (1.1%)**	6 (1.3%)	6 (1.3%)	6 (1.3%)
**Level 4a right**	3 (0.7%)	3 (0.7%)	3 (0.7%)	3 (0.7%)	3 (0.7%)
**Level 6b**	3 (0.7%)	3 (0.7%)	3 (0.7%)	3 (0.7%)	3 (0.7%)
**Level 7a**	**1 (0.2%)**	**2 (0.4%)**	1 (0.2%)	1 (0.2%)	1 (0.2%)
**Level 8 right**	**3 (0.7%)**	**2 (0.4%)**	3 (0.7%)	3 (0.7%)	3 (0.7%)
**Level 7b right**	**1 (0.2%)**	**0 (0.0%)**	1 (0.2%)	1 (0.2%)	1 (0.2%)
**Level 1a**	0 (0.0%)	0 (0.0%)	0 (0.0%)	0 (0.0%)	0 (0.0%)
**Level 4b left**	0 (0.0%)	0 (0.0%)	0 (0.0%)	0 (0.0%)	0 (0.0%)
**Level 4b right**	0 (0.0%)	0 (0.0%)	0 (0.0%)	0 (0.0%)	0 (0.0%)
**Level 6a**	0 (0.0%)	0 (0.0%)	0 (0.0%)	0 (0.0%)	0 (0.0%)
**Level 7b left**	0 (0.0%)	0 (0.0%)	0 (0.0%)	0 (0.0%)	0 (0.0%)

Differences between deep-learning-based automatic assessment and expert correction are highlighted in bold.

## Data Availability

We share the program code and model weights at https://github.com/putzfn/. The raw data are available upon request. However, legal restrictions, especially the EU General Data Protection Regulation (GDPR), the German Data Protection Laws, and the Bavarian Hospital law apply, so some requests may have to be declined partially or completely.

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
