# Peer review of "Deep Learning and Registration-Based Mapping for Analyzing the Distribution of Nodal Metastases in Head and Neck Cancer Cohorts: Informing Optimal Radiotherapy Target Volume Design"

_cancers, 2023, doi:10.3390/cancers15184620_

Round 1
Reviewer 1 Report (Previous Reviewer 2)
Dear members of the editorial board,
The authors amended the manuscript according to the previous recommendations. I would endorse its publication.
Kind regards
Author Response
We want to thank the reviewer for the time for reviewing our manuscript and the very favorable comments.
Reviewer 2 Report (New Reviewer)
Dear authors
congrats very interesting and clinically useful study.
one question: maybe I did not find this information but I would like to gain more information on the time matter: in particular, you presented very well that physicians with more clinical experience, as number 2, show a higher, or better said 100% accuracy compared to physician #3 and #1. When introducing your system the accuracy increases to 100% - thats fantastic but my question is what is the time difference or better time loss when using your automated system?
Author Response
We want to thank the reviewer for the favorable assessment of our work.
The time required for the deep learning-based lymph node level classification was 4 h and 40 min for all 193 datasets in this study. This corresponds to an average of 1 min and 27 s per dataset. The deep learning autosegmentation takes the largest fraction of the required computation time, while the additional computations are very fast. As the method is completely automated, no expert-interaction time is required. If an expert wants to view the predicted classifications for a specific CT dataset, the calculations can be performed in advance on the respective dataset without requiring additional expert time. Comparing the computation time of the automated method with the manual review time was outside the scope of this work, as the computation is automated and does not require user interaction. However, during our evaluation, we experienced that the manual review time was longer than the calculation time. The calculation time can be further accelerated by using a newer GPU generation for deep learning autosegmentation.
The following changes were made to the manuscript to better report the time required for the deep learning automated method:
In the results section the average time required per CT dataset was added:
"[...]. computation for the whole 193-patient cohort by the deep learning method took 4 h and 40 min on a GPU workstation (average of 1 min 27s per CT dataset). [...]"
The following paragraph was added to the discussion section to discuss the time requirements for the deep learning lymph node level classification method:
"[...] computation time for the entire 193-patient cohort took just 4 h and 40 min for the deep learning-based method. This corresponds to an average of 1 min and 27 s per dataset. Comparing the computation time of the automated method with the manual review time was outside the scope of this work. However, as the method is automated, no expert-interaction time is required and if expert validation of predicted classifications is desired, the calculations can be performed in advance. Moreover, the calculation time can be further accelerated by using newer GPU generations for deep learning autosegmentation. As the method scales linearly with the number of datasets, it is particularly suitable for large, multi-center datasets, where manual review alone would be either prohibitively costly or entirely unfeasible. "
Thank you again for the time for reviewing our manuscript and the very favorable comments.
Reviewer 3 Report (New Reviewer)
Congratulations for this fine work.
Minor editing of English language required
Author Response
We want to thank the reviewer for the very favorable assessment of our work. The language was checked by an English native coauthor (SL) following the recommendation and corrected in some areas. The corrections are highlighted in the provided changes-tracked version of the manuscript.
Thank you again for the time for reviewing our manuscript and the very favorable comments.
This manuscript is a resubmission of an earlier submission. The following is a list of the peer review reports and author responses from that submission.
Round 1
Reviewer 1 Report
Weissman et al. detail two of computational methods for analyzing CT-based nodal metastasis distribution in head and neck cancer patients. There is significant utility in the authors’ methods; but, the manuscript is hamstrung by issues that prevent publication at this time.
Chief amongst these issues is the framing of the manuscript. This is essentially a confirmatory study of their prior methods (doi:10.3389/fonc.2023.1115258)—but with a larger cohort and another mapping method previously used in another context (prostate cancer). Focusing the manuscript on these methodological findings (with some modifications) would be key moving forward. The best way to explain this issue (with regards to framing) is by focusing on the authors’ self-described contributions:
“1) describing and evaluating a novel deep learning-based method for assigning lymph node metastases in a 193-patient cohort to one of 20 nodal levels, achieving 98.9% correct automatic assessment as determined by expert review”
- This method is not entirely novel as it was mainly described in the prior publication
- The choice of nodal levels is curious and doesn’t entirely mimic head and neck oncologic practice
o Why segment level 4 but not 2 or 5?
- While 193 patients in-of-itself is a reasonable cohort size; over a 6-year period, it seems like a small number. How were these patients chosen? Does it include all patients that underwent definitive chemoradiation? Is it patients with the same pre-operative imaging? Are they sequential/consecutive patients? More details are needed to explain the small size of the cohort (~30/year).
- Is 3 mm slice thickness (which seems rather thick) standard at your institution? Could this introduce artifact or fill-in with algorithms? Could you comment on this in the manuscript?
- Who was the expert reviewer? If one of the authors, please identify. Please discuss what sort of clinical expertise (and years) that this reviewer has to warrant their being the sole reviewer. This is a critical weakness in the manuscript. One would expect several blinded reviewers; preferably radiologists with identification of their training.
“2) verifying the deep learning method with a novel, complementary non-rigid registration-based mapping technique to a common template, demonstrating consistency between both methods and allowing for analysis and visualization of the nodal metastases distribution of the whole cohort independent of level boundaries”
- This may be the most significant addition to the literature from this manuscript; the authors should focus on this finding and the implications.
“3) revealing considerable spatial inhomogeneity in nodal metastasis distribution with clustering to specific levels and high-risk regions, which could support volume de-escalation approaches and spatially varying, risk-adapted dose prescription in elective neck radiotherapy.”
- The reasoning for this contribution is flawed because of the variety of malignancies/primary cancer sites. One cannot reasonably expect some sort of uniform nodal pattern when cancers from all over the oral cavity, pharynx, and larynx are lumped together. This should be entirely removed from the manuscript as it contradicts much established clinical knowledge and makes sweeping non-data-based presumptions. For example, sparing level 5 (in a patient with nasopharyngeal cancer) may not be wise.
Other points:
- There is nothing straightforward about the Simple Summary; it must be re-written.
- Stating that the analysis took 4h40m may be misleading. Does this including all the time for acquiring the DICOMs, etc. on the machine?
- Remove comparisons to other imaging modalities; that is not the point of this paper
- Tone down all conclusions; focus on specific findings from methods
Reviewer 2 Report
Dear members of the editorial board,
I read this manuscript with great interest. It is well written and methodologically sound. I personally find some parts regarding the performance of other imaging modalities in the Discussion section quite redundant and out of the scope of this paper and would kindly advise the authors to remove or shorten them for the sake of brevity. Because, the current manuscript does not really compare the performance of any diagnostic tools or does not harbor pathology-based information as ground truth etc.
Nevertheless, I would endorse the publication of this manuscript.
Kind regards